# Effect of Low-Level Tragus Stimulation on Cardiac Metabolism in Heart Failure with Preserved Ejection Fraction: A Transcriptomics-Based Analysis

**DOI:** 10.3390/ijms25084312

**Published:** 2024-04-13

**Authors:** Praloy Chakraborty, Monika Niewiadomska, Kassem Farhat, Lynsie Morris, Seabrook Whyte, Kenneth M. Humphries, Stavros Stavrakis

**Affiliations:** 1Heart Rhythm Institute, University of Oklahoma Health Sciences Center, 800 Stanton L Young Blvd, Suite 5400, Oklahoma City, OK 73104, USA; praloy.chakraborty@uhn.ca (P.C.); kassem.farhat@yale.edu (K.F.); lynsie-morris@ouhsc.edu (L.M.); seabrook-whyte@ouhsc.edu (S.W.); 2Peter Munk Cardiac Center, Toronto General Hospital, University Health Network, Toronto, ON M5G 2N2, Canada; 3Aging and Metabolism Research Program, Oklahoma Medical Research Foundation, Oklahoma City, OK 73104, USA; kenneth-humphries@omrf.org

**Keywords:** metabolism, heart failure, tragus stimulation, metabolic regulation

## Abstract

Abnormal cardiac metabolism precedes and contributes to structural changes in heart failure. Low-level tragus stimulation (LLTS) can attenuate structural remodeling in heart failure with preserved ejection fraction (HFpEF). The role of LLTS on cardiac metabolism is not known. *Dahl salt-sensitive rats* of 7 weeks of age were randomized into three groups: low salt (0.3% NaCl) diet (control group; *n* = 6), high salt diet (8% NaCl) with either LLTS (active group; *n* = 8), or sham stimulation (sham group; *n* = 5). Both active and sham groups received the high salt diet for 10 weeks with active LLTS or sham stimulation (20 Hz, 2 mA, 0.2 ms) for 30 min daily for the last 4 weeks. At the endpoint, left ventricular tissue was used for RNA sequencing and transcriptomic analysis. The Ingenuity Pathway Analysis tool (IPA) was used to identify canonical metabolic pathways and upstream regulators. Principal component analysis demonstrated overlapping expression of important metabolic genes between the LLTS, and control groups compared to the sham group. Canonical metabolic pathway analysis showed downregulation of the oxidative phosphorylation (Z-score: −4.707, control vs. sham) in HFpEF and LLTS improved the oxidative phosphorylation (Z-score = −2.309, active vs. sham). HFpEF was associated with the abnormalities of metabolic upstream regulators, including PPARGC1α, insulin receptor signaling, PPARα, PPARδ, PPARGC1β, the fatty acid transporter *SLC27A2*, and lysine-specific demethylase 5A (KDM5A). LLTS attenuated abnormal insulin receptor and *KDM5A* signaling. HFpEF is associated with abnormal cardiac metabolism. LLTS, by modulating the functioning of crucial upstream regulators, improves cardiac metabolism and mitochondrial oxidative phosphorylation.

## 1. Introduction

Heart failure (HF) is the most common cardiovascular cause of hospitalization and mortality [1]. Although HF with preserved ejection fraction (HFpEF) contributes to approximately 50% of patients with HF diagnosis, the pathophysiology of this entity is incompletely understood, and diagnosis and management remain challenging [2]. The pathogenesis of HFpEF is multifactorial, with abnormal diastolic calcium clearance, left ventricular hypertrophy (LVH), and fibrosis leading to diastolic dysfunction of the stiff ventricle. Cardiac inflammation is known to promote cardiac fibrosis and ventricular dysfunction. Unfortunately, there are limited therapeutic options for HFpEF, and many conventional therapeutic strategies for heart failure with reduced ejection fraction (HFrEF) are not successful in HFpEF [2]. Low-level transcutaneous vagus nerve stimulation by electrical stimulation of the tragus of the external ear (LLTS), a noninvasive modality for low-level vagus nerve stimulation (LLVNS), is emerging as a novel nonpharmacological therapeutic option in HFpEF [3]. Both sympatholytic and parasympathomimetic effects of LLVNS are reported to be beneficial [4]. Preclinical as well as clinical studies have demonstrated that LLTS can improve myocardial function and quality of life through amelioration of cardiac inflammation, fibrosis, and hypertrophy [3,5,6].

Abnormal cardiac metabolism or metabolic remodeling plays an essential role in the pathogenesis of HF [7,8]. Metabolic remodelings are known to be present before the onset of abnormal cardiac function and structural remodeling. Aberrant cardiac metabolism results in abnormal energy homeostasis, accumulation of metabolic intermediates, and increased reactive oxygen species (ROS) production, which ultimately leads to mechanical dysfunction, cell death, inflammation, and progressive structural changes [8]. Hence, mitigation of metabolic remodeling may have beneficial effects on HF management. Although LLTS is known to ameliorate cardiac inflammation, fibrosis, and hypertrophy, the role of LLTS on cardiac metabolic remodeling has not been evaluated.

Regulation of expression of genes related to proteins involved in metabolic pathways is one major mechanism of metabolic control, and analysis of gene expression data has been used to identify metabolic alterations in HF [9]. Currently, available high throughput analysis allows simultaneous quantification of the expression pattern of multiple genes in tissue. Further, the knowledge-based bioinformatic algorithms may identify the pathophysiologically altered pathways and elucidate the upstream molecular mechanisms of the alterations based on analysis of differentially expressed genes (DEGs) [10]. The aim of the current study was to evaluate the role of LLTS on cardiac metabolism in HFpEF through ingenuity pathway analysis (IPA) based assessment of differential expression of metabolically important genes in cardiac tissue.

## 2. Results

### 2.1. Differentially Expressed Genes

#### 2.1.1. Effect on Cardiac Phenotype

As described previously, HS diet rats (sham group) developed hypertension, left ventricular hypertrophy, and left ventricular diastolic dysfunction compared to the LS diet group (control group) [5]. HS diet also induced inflammatory cell infiltration and fibrosis in ventricular tissue. Four weeks of LLTS (active group) was associated with significant attenuation of cardiac inflammation and fibrosis, which were translated into mitigation of ventricular hypertrophy and diastolic dysfunction compared to the sham group [5,6]. The effects of LLTS on cardiac remodeling were independent of its antihypertensive effects [5].

#### 2.1.2. Control vs. Sham

A total of eight hundred twenty-six genes (826) were differentially expressed, with three hundred thirty-two (332) genes demonstrating downregulation and four hundred ninety-four (494) demonstrating upregulation in HFpEF tissue. Among the DEGs, 219 genes were involved in metabolism. Other DEGs principally involved inflammatory and profibrotic signaling pathways.

#### 2.1.3. Active vs. Sham

Two hundred ninety-six (296) genes were differentially expressed with downregulation and upregulation of one hundred seventy-three (173) and one hundred twenty-three (123) genes, respectively, in the sham group. Nineteen DEGs belong to metabolic pathways. Again, the majority of other DEGs were members of inflammatory and profibrotic signaling pathways.

### 2.2. Canonical Pathway Analysis

#### 2.2.1. Control vs. Sham

IPA identified nine canonical metabolic pathways and two metabolic signaling pathways as significantly affected (Z-score > 2.0 or <−2.0) (Figure 1, Appendix A). The prediction was based on differential expression of 95 genes and PCA plots demonstrated distinct clustering of those DEGs between control vs. sham groups (Figure 2). All identified metabolic pathways were predicted to be inhibited in the HFpEF group (Figure 1). The inhibition score was highest for the oxidative phosphorylation pathway (Z-score: −4.707). Other altered metabolic pathways included valine degradation (Z-score: −3.00), glutaryl CoA degradation (Z-score: −2.449), fatty acid β oxidation I (Z-score: −3.000), tryptophan degradation III (Z-score: −2.449), isoleucine degradation (Z-score: −2.236), TCA cycle II (Z-score: −2.646), superpathway of geranylgeranyl diphosphate biosynthesis I (via mevalonate) (Z-score: −2.00), and stearate biosynthesis I (Z-score: −2.236). The downregulated genes included important proteins and enzymes for fatty acid transport (*SLC27A1* and *ACSL1*), fatty acid oxidation (*HADHA*&*B* and *ACAA2*), and oxidation, acetyl CoA synthesis from ketone bodies and amino acids (*ACAT1* and *GCDH*), TCA cycle (*IDH3A*), branched-chain amino acid (BCAA) catabolism (*BCAT2, BCKDHA*, and *BCKDHB*), electron transport chain (ETC: complex I, III, IV, and V), and ATP synthesis (*ATP5MF, ATP5PB, ATP5PD*, and *ATP5PO*) (Appendix A). The pathway analysis predicted reduced ATP synthesis (Figure 3) and decreased generation of acetyl-CoA, propionyl-CoA (Figure 4 and Appendix A), and TCA cycle intermediates (Appendix A).

Disrupted metabolic and signaling pathways (*p* < 0.05 and Z-score value > 2) in (A): HFpEF group (LS vs. HS sham) and (B): HS sham group compared to the HS active (LLTS) group.

Pathways selected for analysis: metabolic pathways, mitochondrial dysfunction, and signaling related to metabolism (HIF1α signaling, AMPK, insulin receptor, PI3K/AKT, sirtuin signaling, PPAR signaling, and PPAR α/RXR α activation). Positive Z-score indicates activation, and negative Z-score indicates inhibition in the HS sham group in (A) and (B).

Among the metabolic signaling pathways, sirtuin signaling demonstrated activation (Z-score: 3.900) whereas PPARα/RXRα activation pathway demonstrated inhibition (Z-score: −2.496) in the HS sham group (Figure 1). The differentially expressed metabolically important genes in the sirtuin signaling pathway included the following: *ACADL, ATP5PB, BPGM, GADD45A, GOT2, LDHD, NDUFA4, NDUFA5, NDUFA11, NDUFAB1, NDUFB3, NDUFB4, NDUFB5, NDUFB8, NDUFB10, NDUFS1, NDUFS4, NDUFS6, NDUFS8, NDUFV2, PDK1, PGAM1,* and *PGAM2*. All genes demonstrated downregulation except BPGM and PGAM1, which showed upregulation (Appendix A). Importantly, all genes involved in mitochondrial oxidative phosphorylation (*ATP5PB, NDUFA4, NDUFA5, NDUFA11, NDUFAB1, NDUFB3, NDUFB4, NDUFB5, NDUFB8, NDUFB10, NDUFS1, NDUFS4, NDUFS6, NDUFS8,* and *NDUFV2*) demonstrated down regulation. Through the pathway prediction tool, the activation of the sirtuin pathway in HFpEF was predicted to be associated with reduced fatty acid oxidation, reduced generation of acetyl CoA and TCA cycle intermediates, inhibition of mitochondrial oxidative phosphorylation, and reduced ATP generation (Appendix A). Inhibition of PPARα/RXRα signaling was predicted to be associated with the inhibition of fatty acid oxidation and abnormal glucose homeostasis (Appendix A).

#### 2.2.2. Active vs. Sham

PCA of 95 DEGs, which significantly altered the canonical metabolic pathways and metabolic signaling pathways in HFpEF, also demonstrated an overlapping between the active and control groups, and both these groups were distinct from the sham group (Figure 2). However, only oxidative phosphorylation was found to be significantly modified in IPA prediction with predicted inhibition in the HS sham group compared to the HS active group (Z-score = −2.309). Twelve genes in the mitochondrial oxidative phosphorylation pathway were differentially expressed, and all genes demonstrated a downregulation in the HS sham group (Appendix A). A comparison of the predicted expression pattern of ETC and oxidative phosphorylation proteins suggests that reduced ATP generation in HFpEF is significantly mitigated by LLTS. (Figure 4).

### 2.3. Upstream Regulator Analysis

#### 2.3.1. Control vs. Sham

IPA upstream regulator analysis identified 1697 enriched upstream molecules (genes, RNAs, and proteins; *p* < 0.05). Among those enriched molecules, 91 and 171 upstream regulators demonstrated significant inhibition (Z-score < −2.0) and activation (Z-score > 2.0), respectively, with HFpEF. Among the top 15 inhibited upstream regulators, PPARGC1α (rank 1; Z-score = −5.756), insulin receptor (rank 2; Z-score = −5.55), PPARα (Rank 6; Z-score = −3.778), PPARδ (Rank 13; Z-score = −3.236), and PPARGC1β (rank 15; Z-score = −3.074) were found to play a significant role in cardiac metabolism (Appendix A). In the list of top 15 activated upstream molecules, regulators, SLC27A2 (Rank 10; Z-score = 3.674), and KDM5A (rank 13; Z-score = 3.528) were metabolically important (Appendix A).

#### 2.3.2. Active vs. Sham

Among 371 enriched upstream regulators (*p*-value of overlap < 0.05), 4 molecules demonstrated significant inhibition (Z-score < −2.0), and 5 molecules demonstrated inhibition (Z-score > 2.0) in the sham group (Appendix A). Insulin receptor (INSR, Z-score: −2.00; rank 4) was in the list of inhibited molecules, whereas and KDM5A (rank 3; Z-score: 2.530) was an activated molecule.

## 3. Discussion

In the current study, we used IPA-based analysis of next-generation RNA sequence data to predict the disrupted metabolic pathways and regulatory molecules behind those metabolic alterations from DEGs. Our genomic-based data suggested that (i) HFpEF is associated with abnormal mitochondrial energy synthesis as well as abnormal substrate metabolism. (ii) Although canonical pathway analysis indicated improvement solely in mitochondrial oxidative phosphorylation with LLTS, overlapping of differentially expressed genes implicated in critical cardiac metabolism and metabolic signaling between the control and LLTS group, compared to the sham group, on PCA plots suggests that LLTS in HFpEF tends to restore the expression of important metabolic genes towards normal level. (iii) Inhibition of PPAR and insulin receptor signaling, as well as activation of sirtuin and KDM5A signaling, may be responsible for metabolic dysregulations. LLTS mitigated the abnormalities in insulin and KDM5A signaling. Metabolic maladaptations in heart failure may result from sympathovagal imbalance [11]. LLTS is shown to exert cholinomimetic and antiadrenergic action on the heart, and both these mechanisms may contribute to metabolic benefits [3].

Mitochondrial oxidative phosphorylation was the most disrupted metabolic pathway in our HFpEF model. Inhibition of mitochondrial phosphorylation with subsequent reduced ATP generation was predicted. The normal heart principally relies on oxidative metabolism, and 95% of ATP is generated by mitochondrial oxidative phosphorylation [8]. Abnormal mitochondrial function and reduced ATP generation are the earliest manifestations of HF [8,12,13]. Malfunction of the ATP-dependent cardiac sarco/endoplasmic reticulum Ca^2+^-ATPase (SERCA2a) due to reduced cellular ATP pool may explain impaired diastolic Ca^2+^ clearance and ventricular diastolic dysfunction in the early stages of HF [13].

Enhanced expression of mitochondrial genes by LLTS with the predicted improvement of ATP generation is a novel finding of our study. Vagus nerve stimulation (VNS) is shown to improve mitochondrial biogenesis function with subsequent improvement of ATP degeneration in ventricular cardiomyocytes following ischemic and nonischemic insults [14,15]. Muscarinic cholinergic stimulation is shown to improve mitochondrial function [14]. The cholinomimetic effect of LLTS may contribute to the improvement of mitochondrial function.

The current study also suggested reduced generation of acetyl-CoA from FA, ketone bodies, and amino acids in HFpEF. Inhibition of the TCA cycle was also predicted. FA’s are the dominant fuel of the heart. However, glucose and ketone bodies play crucial roles in the setting of metabolic challenges [8]. Acetyl CoA, generated from oxidative metabolism of all the above substrates (FA, glucose, ketoacids, and amino acids), is utilized by the TCA cycle to generate reduced nicotinamide adenine dinucleotide (NADH) that is oxidized by ETC to produce ATP [8]. Although myocardial substrate utilization varies according to etiology (ischemic vs. nonischemic), type (HFpEF vs. HFrEF), and stages of HF, other than deranged FAO, use of alternative fuels such as glucose, ketones, BCAAs, and TCA cycle intermediates are also compromised in HFpEF, indicating insufficient anaplerosis [8]. Both valine and isoleucine belong to BCAAs [7,16]. Although BCAA catabolism provides a nonsignificant contribution as a myocardial substrate, cardiac BCAA clearance plays an important role in maintaining serum BCAA levels. An increase in serum BCAA due to impaired cardiac clearance is reported to contribute to insulin resistance and cardiac fibrosis in HF [17,18]. In addition, BCAAs have an important role in cardiac physiology due to their ability to activate mTOR signaling [19]. Although VNS is shown to modulate myocardial substrate metabolism [14], LLTS did not alter myocardial substrate metabolism pathways in our study. The differences in experimental models (ischemic vs. nonischemic), type of HF (HFpEF vs. HFrEF), techniques used (invasive vs. noninvasive vagus nerve stimulation), and duration of treatment may be responsible for this difference.

Analysis of canonical signaling pathways and upstream regulator analysis in our study demonstrated inhibition of PPAR and insulin signaling and upregulation of sirtuin signaling and KMD5A in HFpEF tissue. The peroxisome proliferator-activated receptors (PPAR) are transcription factors that regulate the expression of proteins involved in metabolic homeostasis. The PPAR axis also mitigates inflammation and cardiac structural remodeling [20]. PPARα promotes FAO by inducing the expression of protein fatty acid transport and mitochondrial and paroxysmal fatty acid oxidation. PPARδ, in addition to FAO, promotes the oxidative metabolism of glucose [20]. The proper functioning of PPAR requires interaction with the retinoid X receptor (RXR). PPARα, in association with PPARγ-coactivator 1α (PGC1α), also promotes mitochondrial biogenesis and function [12,20]. In our study, canonical pathway analysis demonstrated inhibition of PPARα/RXRα, and upstream regulator analysis demonstrated significant inhibition of PGC-1α, PPARα, and PPARδ. A combination of gene and protein expression analysis reported abnormal PPAR signaling in a rodent model of HFpEF [21]. Abnormal PPAR signaling is extensively linked to abnormal substrate metabolism, reduced mitochondrial biogenesis, mitochondrial dysfunction, and energy deficiency in HF [20,21].

Sirtuin proteins (Sirt1–7) are nicotinamide adenine dinucleotide (NAD) dependent histone deacetylases that regulate cardiac metabolism as well as cell survival and growth [22]. Sirtuin proteins regulate mitochondrial function by deacetylating nuclear and mitochondrial proteins as well as other metabolic signaling molecules like PGC-1α, PPAR-α, and Akt/PI3K pathways [12,22]. In our HFpEF model, despite the activation of sirtuin signaling, reduced expression of mitochondrial genes was noted. Hence, the upregulation of sirtuin signaling may be a compensatory response to mitochondrial dysfunction. However, a nonphysiological activation of sirtuin signaling is also reported to exacerbate ventricular dysfunction by inhibiting the expression of mitochondrial genes and thus inducing mitochondrial dysfunction [22,23]. The activated sirtuin signaling may also play a detrimental role by promoting cardiomyocyte growth and hypertrophy [24].

Insulin signaling is another important regulator of cardiomyocyte glucose uptake and metabolism. Reduced insulin action due to insulin resistance is linked to abnormal cardiac metabolism in HF [8]. Insulin resistance is reported to be present with HFpEF even before the onset of diabetes mellitus, a systemic metabolic manifestation of insulin resistance [25]. Mitigation of abnormal insulin signaling in the active LLTS group is a novel finding of our study. Insulin resistance in HF is linked to chronic sympathoexcitation [26]. Defective cholinergic function is also associated with cardiac metabolic dysfunction in diabetes, and improvement of the cardiac cholinergic system has been found to ameliorate cardiac metabolic abnormalities in diabetic cardiomyopathy [27]. Hence, both antiadrenergic and parasympathomimetic effects may contribute to the mitigation of insulin receptor dysfunction in our study.

Activation of KMD5A signaling in HFpEF and inhibition of KMD5A signaling by LLTS are other novel findings of our study. KMD5A, a member of histone lysine demethylases, modulates gene expression by epigenetic modification [28]. During the maturation of cardiomyocytes, the switching from nonoxidative to oxidative metabolism is carried out by activating histone modification through trimethylation of histone H3 at lysine residue 4 (H3K4me3) [29]. KMD5A is shown to repress the expression of genes associated with oxidative phosphorylation (OXPHOS) and FAO [30]. Inhibition of KMD5A resulted in the upregulation of FAO and OXPHOS genes in the human pluripotent stem cell-derived cardiomyocyte (iPSC-CMs) model, and this gene programming was translated into improved myocardial calcium handling [30]. Hence, inhibition of KMD5A by LLTS may contribute to improved mitochondrial oxidative phosphorylation in our study.

## 4. Materials and Methods

### 4.1. Study Protocol

The study was conducted as per the guidelines of the animal research committee, and the protocol was approved by the University of Oklahoma Health Sciences Center Institutional Animal Care and Use Committee. Ventricular tissues were obtained from our previous study, and the protocol is described in detail in his paper [5]. In brief, dahl salt-sensitive (DS) rats (Charles River Laboratories, Wilmington, MA, USA) were fed 0.3% salt (NaCl) until 7 weeks of age. After the completion of 7 weeks, one group (*n* = 6) was continued with 0.3% NaCl (low salt or LS), whereas another group was fed 8% NaCl (high salt or HS group) for the next 6 weeks. After 13 weeks, the HS group was randomized to active LLTS (*n* = 8) and sham (*n* = 5) groups (Figure 5). Active and sham groups received transcutaneous electrical stimulation through a nerve stimulation device (InTENSity Twin Stim; Current Solutions LLC, Austin, TX, USA) for the next 4 weeks. The stimulation was delivered daily for 30 min under isoflurane (2%) with the following parameters: 20-Hz frequency, 0.2-ms pulse duration, and 3-mA amplitude. In the active LLTS group, the electrodes were placed over the auricular concha region with the cathode inside and the anode outside, as previously described [6]. In the sham group, electrodes were placed over the tip of the auricular margin.

### 4.2. Next-Generation RNA Sequencing

At the end of 17 weeks, rats were sacrificed, and cardiac tissue was collected for transcriptomic analysis. In brief, LV tissue was homogenized, and RNA was isolated using RNeasy Mini Kit and QIAzol Lysis Reagent (Qiagen Sciences, Inc., Germantown, MD, USA). Subsequently, RNA sequencing analysis was performed using an Illumina NovSeq 6000 system, and libraries were constructed using NEBNext Ultra II Library Prep Kit (New England Biolabs, Inc., Ipswich, MA, USA) as per established protocols. The HISAT2 software(version 2.2.1) was used to align the RNA sequencing data with the reference genome (Rattus norvegicus rn6). The transcripts were filtered (at least 0.5 cpm present in each sample per group), yielding a total of 12,979 transcripts, and normalized for library depth [31]. A list of DEGs was created with the help of the limma-voom procedure in Partek Flow 10.0.23.0131 (Partek, Inc., St. Louis, MO, USA) [32].

### 4.3. Ingenuity Pathway Analysis

The Ingenuity Pathway Analysis tool (IPA^®^, QIAGEN version 2023.1, RedwoodCity, www.qiagen.com/ingenuity; accessed between 10–12 June 2023) was used to identify canonical pathways and upstream regulators from DEGs. The LS control and HS sham group (control vs. sham) comparison represented the altered pathways and upstream regulators with HFpEF, and the HS-LLTS vs. HS sham (active vs. sham) comparison represented the effect of LLTS on HFpEF. Our study was focused on the evaluation of cardiac metabolism, mitochondrial function, and signaling pathways known to be involved in metabolic regulation. Other than metabolic pathways, canonical pathways were selected from cellular stress and injury signaling (HIF1α signaling), ingenuity toxicity list pathway (mitochondrial dysfunction), intracellular and second messenger signaling (AMPK, insulin receptor, PI3K/AKT, and sirtuin signaling), and nuclear receptor signaling (PPAR signaling and PPAR α/RXR α activation) for analysis. In addition to activity analysis, the pathway analysis tool was also used to predict the functional effects of significantly enriched pathways. IPA upstream analysis was focused on identifying the genes, RNA, and proteins involved in metabolic gene regulation.

### 4.4. Statistical Analysis

Prior to the analysis of canonical pathways and upstream regulators, raw gene expression data were preprocessed to filter out genes that did not meet the specified criteria, ensuring that only differentially expressed genes (DEGs) between groups (control vs. sham/active vs. sham) were included in the downstream analysis. This preprocessing step aimed to prioritize genes exhibiting biologically meaningful changes in expression levels for further investigation using IPA. DEGs were identified based on statistical significance (using unpaired *t*-test) and fold change. Genes with *p* < 0.05 and >1.5-fold change in expression levels between experimental groups were used for subsequent IPA [32]. The statistical significance of identified pathways and upstream regulators was determined through a series of established methods integrated within the IPA platform. Canonical pathways enriched with DEGs were identified through the enrichment analysis module. This module applies Fisher’s exact test to calculate the statistical significance of pathway enrichment, and –log (*p*-value) > 1.3 was considered as the enrichment threshold [33]. IPA applies a combination of enrichment analysis and causal network analysis to predict upstream regulators based on the direction and magnitude of gene expression changes, and a *p*-value of overlap < 0.05 was considered enriched [34]. Enriched pathways or molecules with activation Z-score beyond 2 (Z-score > 2 or <2) were considered significantly altered. A positive Z-score suggested activation, and a negative suggested inhibition [34]. Results from canonical pathway analysis and upstream regulator analysis were visualized using interactive diagrams and graphs provided by the IPA software. These visualizations aided in the interpretation of statistically significant pathways and regulators, allowing for a comprehensive understanding of the biological processes underlying our experimental findings. To understand the relationship of gene expression profile between cohorts (control, sham, and active), principal component analysis (PCA) plots were generated using R-project (https://scienceinside.shinyapps.io/mvda/, accessed between 11–15 January 2024), and the first two components were used for visualization.

### 4.5. Limitations

To the best of our knowledge, this is the first study that evaluated the effects of noninvasive LLTS on metabolic remodeling in HFpEF. However, we acknowledge the limitations of this study: First, we understand the limitations imposed by the relatively small sample size. However, we believe that this exploratory pilot study provided valuable preliminary insights into the effects of LLTS on cardiac metabolism in HFpEF. Second, the metabolic phenotype of HFpEF differs according to etiology, stages, and species. Hence, results from this study on a high salt-induced rodent model should be translated into human HFpEF with caution. Third, we understand that by analyzing the gene expression data, IPA provides insights into biological pathways and regulators, and the predictions generated by the software are based on existing knowledge curated from literature and databases. However, metabolic pathways are also modulated by trafficking, degradation, posttranslational modifications, and allosteric modulations of proteins. Future studies with simultaneous protein analysis and functional assays will further elucidate the mechanisms. Fourth, in this study, we used one-hour daily stimulation for four weeks. However, metabolic effects may differ depending on the duration of stimulation as well as stimulation parameters. Finally, the detailed molecular mechanisms (sympatholytic vs. parasympathomimetic or both) of metabolic benefits of LLTS were not evaluated in our study.

## 5. Conclusions

IPA analysis of transcriptomics data from cardiac tissue suggests that HFpEF is associated with abnormal substrate and energy metabolism. Abnormal insulin, PPAR, and sirtuin signaling, as well as epigenetic modifications from KMD5A activation, may contribute to metabolic remodeling. LLTS, by improving insulin signaling and inhibition of KMD5A, may improve mitochondrial function and energy homeostasis.

## Figures and Tables

**Figure 1 ijms-25-04312-f001:**
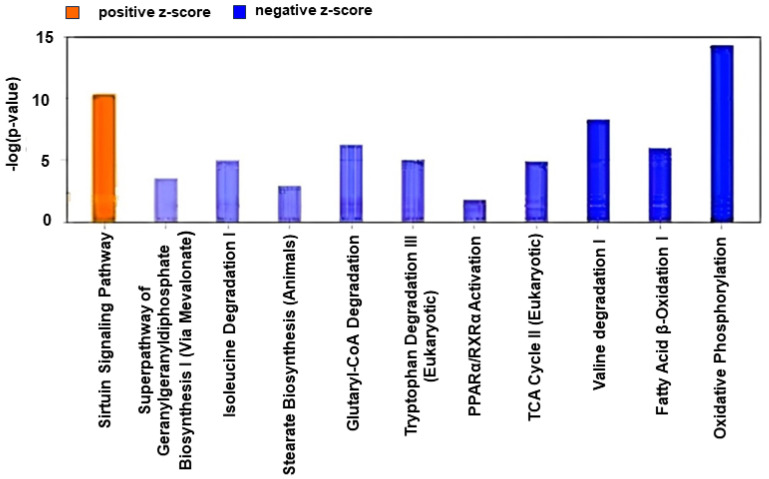
IPA comparison analysis of metabolic and signaling pathways.

**Figure 2 ijms-25-04312-f002:**
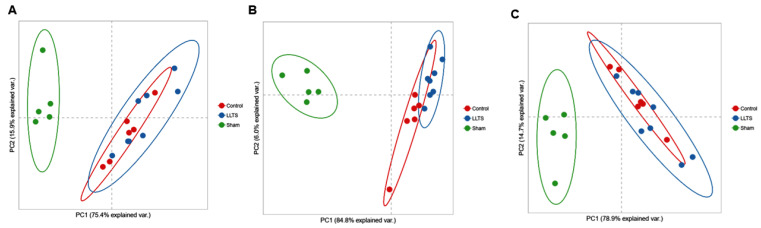
Principal component analysis of differentially expressed genes related to enriched metabolic (**A**), metabolic signaling (**B**), and oxidative phosphorylation pathways (**C**). On the first two principal components, each of the three groups clusters together with overlap between control (LS) and active (LLTS) groups. Sham (HS sham) group clusters further away from control and LLTS. HS: high salt; LLTS: low-level tragus stimulation; LS: low salt.

**Figure 3 ijms-25-04312-f003:**
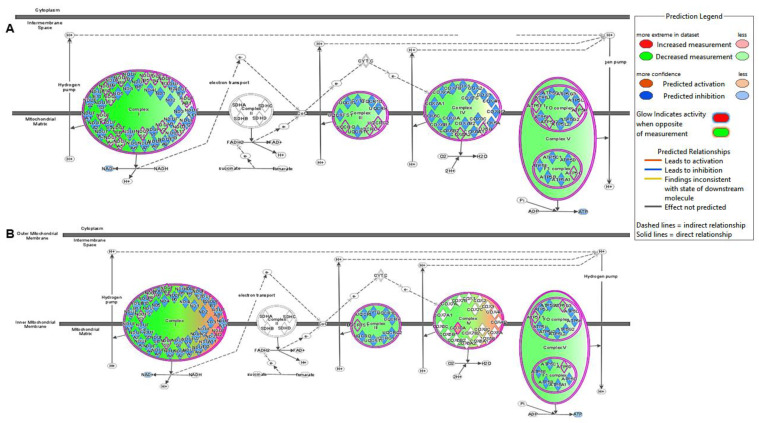
IPA prediction of oxidative phosphorylation pathway in response to LLTS. Comparison between LS vs. HS sham (**A**) and HS sham group compared to HS active (**B**). IPA analysis predicted the downregulation of the oxidative phosphorylation pathway in the HFpEF (HS sham group) compared to the control group, suggesting a potential impairment of mitochondrial function in HFpEF. In contrast, an improvement in the oxidative Phosphorylation pathway was observed in the HS active group following LLTS compared to the HS sham group, indicating a potential beneficial effect of LLTS on mitochondrial function in HFpEF. The predictions were based on transcriptomic analysis data and visualized using interactive diagrams generated by Ingenuity Pathway Analysis (IPA) software. Color coding in the figure represents predicted changes in functional status (activation/inhibition) and quantity of different in different pathways and molecules, respectively. HS: high salt; LLTS: Low-level tragus stimulation; LS: low salt.

**Figure 4 ijms-25-04312-f004:**
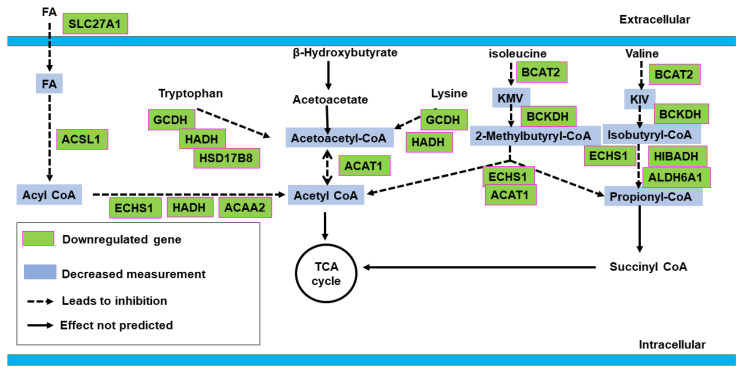
IPA prediction of acetyl-CoA and propionyl-CoA generation (LS vs. HS sham). Acetyl-CoA is generated from FA, ketone bodies, and amino acids. IPA predicted reduced generation of acetyl-CoA from all sources and propionyl-CoA from valine and isoleucine. Abbreviations: ACAA2—acetyl-coenzyme A acyltransferase 2; ACAT1—acetyl-CoA acetyltransferase 1; ACSL1—acyl-CoA synthetase long-chain family member 1; ALDH6A1—aldehyde dehydrogenase 6 family, member A1; BCAT2—branched-chain amino acid transaminase 2; BCKDH—branched-chain α-ketoacid dehydrogenase complex; ECHS1—enoyl-CoA hydratase, short chain 1; FA—fatty acid; GCDH—glutaryl-CoA dehydrogenase; HADH—hydroxyacyl-coenzyme A dehydrogenase; HIBADH—β-hydroxybutyrate dehydrogenase; HSD17B8—hydroxysteroid 17-beta dehydrogenase 8. Decreased measurement indicates Ingenuity Pathway Analysis predicted reduced concentration of metabolic product based on altered gene expression.

**Figure 5 ijms-25-04312-f005:**
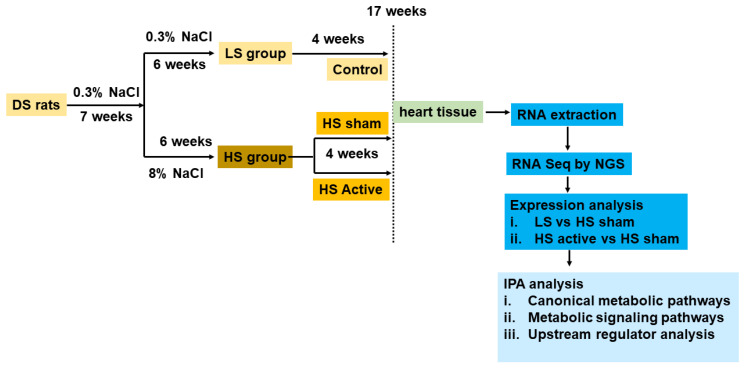
Experimental protocol: HS active indicates the HFpEF group receiving LLTS.

## Data Availability

The data presented in this study are available on request from the corresponding author due to Institute policy.

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
