# Peer review of "Effect of Low-Level Tragus Stimulation on Cardiac Metabolism in Heart Failure with Preserved Ejection Fraction: A Transcriptomics-Based Analysis"

_ijms, 2024, doi:10.3390/ijms25084312_

Round 1

Reviewer 1 Report

Comments and Suggestions for Authors

Manuscript is well written and the research work is well presented. However, I have some minor comments about this work.

First, Authors used low N number for the sham group. Can the authors justify this issue.

Second, authors mentioned that acetyl-CoA levels are decreased in HFpEF. It is well established that acetylation modulates the activity of key metabolic enzymes involved in fatty acid and glucose oxidation. Does the authors evaluate the changes in acetylation in active vs sham groups? 

Reviewer 2 Report

Comments and Suggestions for Authors

The article is innovative and provides valuable information. The statistical analysis should be expanded with additional appropriate tests. At this point, it is too superficial. Improving this aspect is recommended. It is unclear in the results where and which test was applied, for example, the t-test. Results for the t-test should be recorded according to the appropriate standard (e.g., U = 23; p = 0.03). The effect size for the t-test should be calculated using the appropriate measure. The statistical software used was not indicated. Figure 4 is completely unclear. The labels on Figure 1 are illegible. The approval number of the Ethics Committee is missing.

Comments on the Quality of English Language

Minor editing of English language required.
